# Hyperbolic Attention Networks

**Caglar Gulcehre, Misha Denil, Mateusz Malinowski, Ali Razavi,**
**Razvan Pascanu, Karl Moritz Hermann, Peter Battaglia, Victor Bapst,**
**David Raposo, Adam Santoro, Nando de Freitas**

DeepMind

## Abstract

Recent approaches have successfully demonstrated the benefits of learning the parameters of shallow networks in hyperbolic space. We extend this line of work by imposing hyperbolic geometry on the embeddings used to compute the ubiquitous attention mechanisms for different neural networks architectures. By only changing the geometry of embedding of object representations, we can use the embedding space more efficiently without increasing the number of parameters of the model. Mainly as the number of objects grows exponentially for any semantic distance from the query, hyperbolic geometry –as opposed to Euclidean geometry– can encode those objects without having any interference. Our method shows improvements in generalization on neural machine translation on WMT'14 (English to German), learning on graphs (both on synthetic and real-world graph tasks) and visual question answering (CLEVR) tasks while keeping the neural representations compact.

## 1 Introduction

The focus of this work is to endow neural network representations with suitable geometry to capture fundamental properties of data, including hierarchy and clustering behaviour. These properties emerge in many real-world scenarios that approximately follow power-law distributions (Newman, 2005; Clauset et al., 2009). This includes a wide range of natural phenomena in physics (Lin and Tegmark, 2017), biology (McGill et al., 2006), and even human-made structures such as metabolic-mass relationships (Borg, 1982), social networks (Krioukov et al., 2010; Papadopoulos et al., 2010), and frequencies of words (Powers, 1998; Piantadosi, 2014; Takahashi and Tanaka-Ishii, 2017).

Complex networks (Krioukov et al., 2010), which connect distinguishable heterogeneous sets of elements represented as nodes, provide us an intuitive way of understanding these structures. They will also serve as our starting point for introducing hyperbolic geometry, which is by itself difficult to visualize. Nodes in complex networks are referred to as heterogeneous, in the sense that they can be divided into sub-nodes which are themselves distinguishable from each other. The scale-free structure of natural data manifests itself as a power law distribution on the node degrees of the complex network that describes it.

Complex networks can be approximated with tree-like structures, such as taxonomies and dendrograms, and as lucidly presented by Krioukov et al. (2010), hyperbolic spaces can be thought of as smooth trees abstracting the hierarchical organization of complex networks. Let us begin by recalling a simple property of $n$-ary trees that will help us understand hyperbolic space and why hyperbolic geometry is well suited to model relational data.

In an $n$-ary tree, the number of nodes at distance $r$ from the root and the number of nodes at distance no more than $r$ from the root both grow as $n^r$. Similarly, in a two-dimensional hyperbolic space with curvature $-\zeta^2, \zeta > 0$, the circumference and area of a disc of radius $r$ grows as $2\pi\sinh(\zeta r)$ and $2\pi(\cosh(\zeta r) - 1)$, respectively, both of are exponential in $r$ (Krioukov et al., 2009; 2010). The growth of volume in hyperbolic space should be contrasted with Euclidean space where the corresponding quantities expand polynomially, circumference as $2\pi r$ and area as $\pi r^2$.

In the two-dimensional example of Figure 1, the expanding rings show examples at a fixed semantic distance from the central object ("pug"). The number of concepts grows quickly with semantic distance forcing each successive ring to be more crowded in order to maintain a fixed distance to the center. In contrast, the extra volume of hyperbolic spheres (depicted by reducing the size of the examples) allows all of the examples to remain well separated from their semantic neighbours.

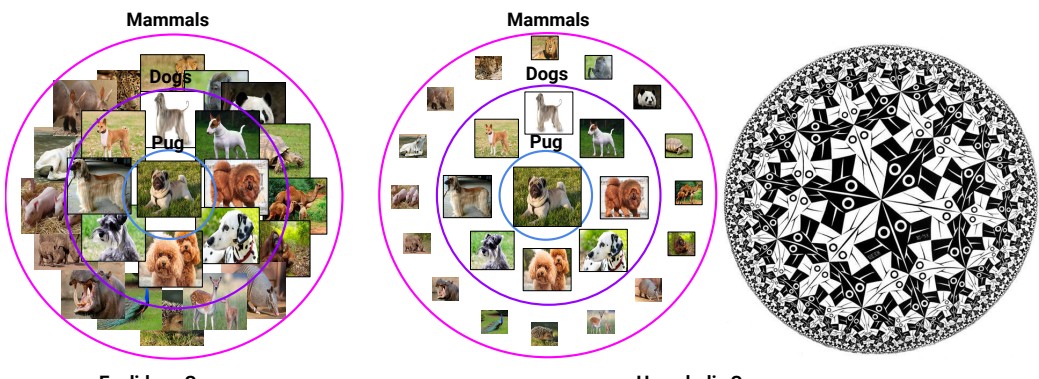

Figure 1: An intuitive depiction of how images might be embedded in 2D. The location of the embeddings reflects the similarity between each image and that of a pug. Since the number of instances within a given semantic distance from the central object grows exponentially, the Euclidean space is not able to compactly represent such structure (left). In hyperbolic space (right) the volume grows exponentially, allowing for sufficient room to embed the images. For visualization, we have shrunk the images in this Euclidean diagram, a trick also used by Escher.

Mechanically, the computed embeddings by a random network for objects at a given semantic distance might still seem epsilon distance away from each other (or crowded) as the ones obtained by using Euclidean geometry. However, enforcing hyperbolic geometry intuitively means that all operations with these embeddings take into account, the density in that particular region of the space. For example, any noise introduced in the system (e.g., in gradients) will also be corrected by the density. In contrast to working in Euclidean space, this means that the embeddings will be equally distinguishable regardless of the density.

The intimate connection between hyperbolic space and scale free networks (where node degree follows a power law) is made more precise in Krioukov et al. (2010). In particular, there it is shown that the heterogeneous topology implies hyperbolic geometry, and conversely hyperbolic geometry yields heterogeneous topology. Moreover, Sarkar (2011) describes a construction that embeds trees in two-dimensional hyperbolic space with arbitrarily low distortion, which is not possible in Euclidean space of any dimension (Linial et al., 1998). Following this exciting line of research, recently the machine learning community has gained interest in learning non-Euclidean embeddings directly from data (Nickel and Kiela, 2017; Chamberlain et al., 2017; Ritter, 1999; Ontrup and Ritter, 2002; Tay et al., 2018; Bronstein et al., 2017).

Fuelled by the desire of increasing the capacity of neural networks without increasing the number of trainable parameters so as to match the complexity of data, we propose *hyperbolic attention networks*. As opposed to previous approaches, which impose hyperbolic geometry on the parameters of shallow networks (Nickel and Kiela, 2017; Chamberlain et al., 2017), we impose hyperbolic geometry on the activations of deep networks. This allows us to exploit hyperbolic geometry to reason about embeddings produced by *deep* networks. We introduce efficient hyperbolic operations to express the popular, ubiquitous mechanism of attention (Bahdanau et al., 2014; Duan et al., 2017; Vaswani et al., 2017; Wang et al., 2017). Our method shows improvements in terms of generalization on neural machine translation (Vaswani et al., 2017), learning on graphs and visual question answering (Antol et al., 2015; Malinowski and Fritz, 2014; Johnson et al., 2017) tasks while keeping the representations compact. Simultaneously to our work, Cho et al. (2018) proposed a method to learn SVMs in the hyperboloid model of hyperbolic space, and Nickel and Kiela (2018) proposed a method to learn shallow embeddings of graphs in hyperbolic space by using the hyperboloid model.

## 2  MODELS OF HYPERBOLIC SPACE

Hyperbolic space cannot be isometrically embedded into Euclidean space (Krioukov et al., 2010); however, there are several ways to endow different *subsets* of Euclidean space with a hyperbolic

metric, leading to different *models* of hyperbolic space. This leads to the well known Poincaré ball model (Iversen, 1992) and many others.

The different models of hyperbolic space are all essentially the same, but different models define different coordinate systems, which offer different affordances for computation. In this paper, we primarily make use of the hyperboloid (or Lorentz) model of the hyperbolic space. Since the hyperboloid is unbounded, it a convenient target for projecting into hyperbolic space. We also make use of the Klein model, because it admits an efficient expression for the hyperbolic aggregation operation we define in Section 4.2.

We briefly review the definitions of the hyperboloid and Klein models and the relationship between them, in just enough detail to support the presentation in the remainder of the paper. A more thorough treatment can be found in Iversen (1992). The geometric relationship between the Klein and hyperboloid models is diagrammed in Figure 5 of the supplementary material.

**Hyperboloid model:** This model of $n$ dimensional hyperbolic space is a manifold in the $n+1$ dimensional Minkowski space. The Minkowski space is $\mathbb{R}^{n+1}$ endowed with the indefinite Minkowski bilinear form

$$\langle \mathbf{q}, \mathbf{k} \rangle_M = \sum_{i=1}^{n} q_i k_i - q_{n+1} k_{n+1}.$$

The hyperboloid model consists of the set

$$\mathbb{H}^n = \{\mathbf{x} \in \mathbb{R}^{n+1} | \langle \mathbf{x}, \mathbf{x} \rangle_M = -1, \, x_{n+1} > 0\}$$

endowed with the distance metric $d_{\mathbb{H}}(\mathbf{q}, \mathbf{k}) = \operatorname{arccosh}(-\langle \mathbf{q}, \mathbf{k} \rangle_M)$.

**Klein model:** This model of hyperbolic space is a subset of $\mathbb{R}^n$ given by $\mathbb{K}^n = \{\mathbf{x} \in \mathbb{R}^n | \|\mathbf{x}\| < 1\}$, and a point in the Klein model can be obtained from the corresponding point in the hyperboloid model by projection

$$\pi_{\mathbb{H} \to \mathbb{K}}(\mathbf{x})_i = \frac{x_i}{x_{n+1}},$$

with its inverse given by

$$\pi_{\mathbb{K} \to \mathbb{H}}(\mathbf{x}) = \frac{1}{\sqrt{1 - \|\mathbf{x}\|^2}} (\mathbf{x}, 1)$$

Distance computations in the Klein model can be inherited from the hyperboloid, in the sense that $d_{\mathbb{K}}(\mathbf{q}, \mathbf{k}) = d_{\mathbb{H}}(\pi_{\mathbb{K} \to \mathbb{H}}(\mathbf{k}), \pi_{\mathbb{K} \to \mathbb{H}}(\mathbf{q}))$.

## 3 ATTENTION AS A BUILDING BLOCK FOR RELATIONAL REASONING

Learning relations in a graph by using neural networks to model the interactions or relations has shown promising results in visual question answering (Santoro et al., 2017), modelling physical dynamics (Battaglia et al., 2016), and reasoning over graphs (Li et al., 2015; Vendrov et al., 2016; Kipf et al., 2018; Kool and Welling, 2018). Graph neural networks (Li et al., 2015; Battaglia et al., 2016) incorporate a message passing as part of the architecture in order to capture the intrinsic relations between entities. Graph convolution networks (Bruna et al., 2013; Kipf and Welling, 2016; Defferrard et al., 2016) use convolutions to efficiently learn a continuous-space representation for a graph of interest.

Many of these relational reasoning models can be expressed in terms of an attentive read operation. In the following subsection, we give a general description of the attentive read, and then discuss its specific instantiations in two relational reasoning models from the literature.

### 3.1 ATTENTIVE READ

First introduced for translation in Bahdanau et al. (2014), attention has seen widespread use in deep learning, not only for applications in NLP but also for image processing (Wang et al., 2017) imitation

learning (Duan et al., 2017) and memory (Graves et al., 2016). The core computation is the *attentive read* operation, which has the following form:

$$\mathbf{r}(\mathbf{q}_i, \{\mathbf{k}_j\}_j) = \sum_j \frac{f(\mathbf{q}_i, \mathbf{k}_j)}{Z} \mathbf{v}_{ij}. \tag{1}$$

Here $\mathbf{q}_i$ is a vector called the *query* and the $\mathbf{k}_j$'s are the *keys* for the memory locations being read from. The pairwise function $f(\cdot, \cdot)$ computes a scalar matching score between a query and a key, and the vector $\mathbf{v}_{ij}$ is a *value* to be read from location $j$ by query $i$. $Z > 0$ is a normalization factor for the full sum. Both $\mathbf{v}_{ij}$ and $Z$ are free to depend on arbitrary information, but we leave any dependencies here implicit.

It will be useful in the discussion to break this operation down into two parts. The first is the **matching**, which computes attention weights $\alpha_{ij} = f(\mathbf{q}_i, \mathbf{k}_j)$ and the second is the **aggregation**, which takes a weighted average of the values using these weights,

$$m_i(\{\alpha_{ij}\}_j, \{\mathbf{v}_{ij}\}_j) = \sum_j \frac{\alpha_{ij}}{Z} \mathbf{v}_{ij}.$$

Instantiating a particular attentive read operation involves specifying both $f(\cdot, \cdot)$ and $\mathbf{v}_{ij}$ along with the normalization constant $Z$.

If one performs an attentive read for each element of the set $j$ then the resulting operation corresponds in a natural way to message passing on a graph, where each node $i$ aggregates messages $\{\mathbf{v}_{ij}\}_j$ from its neighbours along edges of weight $f(\mathbf{q}_i, \mathbf{k}_j)/Z$.

We can express many (although not all) message passing neural network architectures (Gilmer et al., 2017) using the attentive read operation of Equation 1 as a primitive. In the following sections we do this for two architectures and then discuss how we can replace both the matching and aggregation steps with versions that leverage hyperbolic geometry.

## 3.2 RELATION NETWORKS

Relation Networks (RNs) (Santoro et al., 2017) are a neural network architecture designed for reasoning about the relationships between objects. An RN operates on a set of objects $O$ by applying a shared operator to each pair of objects $(\mathbf{o}_i, \mathbf{o}_j) \in O \times O$. The pairs can be augmented by a global information, and the result of each relational operation is passed through a further global transformation.

Using the notation of the previous section, we can write the RN as

$$RN(O, \mathbf{c}) = h\left(\sum_i \mathbf{r}(\mathbf{o}_i, \{\mathbf{o}_j\}_j))\right),$$

where $f(\mathbf{o}_i, \mathbf{o}_j) = 1$, $\mathbf{v}_{ij} = g(\mathbf{o}_i, \mathbf{o}_j, \mathbf{c})$, $Z = 1$. $h$ is the global transformation, $g$ is the local transformation and $\mathbf{c}$ is the global context, as described in Santoro et al. (2017). We augment the basic RN to allow $f(\mathbf{o}_i, \mathbf{o}_j) \in [0, 1]$ to be a general learnable function.

Interpreting the RN as learned message passing on a graph over objects, the attention weights take on the semantics of edge weights, where $\alpha_{ij}$ can be thought of as the probability of the (directed) edge $\mathbf{o}_j \to \mathbf{o}_i$ appearing in the underlying reasoning graph.

## 3.3 SCALED DOT-PRODUCT ATTENTION

In the Transformer model of Vaswani et al. (2017) the authors define an all-to-all message passing operation on a set of vectors which they call scaled dot-product attention. In the language of Section 3.1 the scaled dot-product attention operation performs several attentive reads in parallel, one for each element of the input set.

Vaswani et al. (2017) write scaled dot-product attention as $\mathbf{R} = \mathrm{softmax}\left(\frac{\mathbf{Q}\mathbf{K}^T}{\sqrt{d}}\right)\mathbf{V}$, where $\mathbf{Q}$, $\mathbf{K}$ and $\mathbf{V}$ are referred to as the *queries*, *keys*, and *values* respectively, and $d$ is the shared dimensionality of the queries and keys. Using lowercase letters to denote rows of the corresponding matrices, we can write each row of $\mathbf{R}$ as the result of an attentive read with

$$f(\mathbf{q}_i, \mathbf{k}_j) = \exp\left(\frac{1}{\sqrt{d}}\langle \mathbf{q}_i, \mathbf{k}_j \rangle\right), \qquad \mathbf{v}_{ij} = \mathbf{v}_j, \qquad Z = \sum_j f(\mathbf{q}_i, \mathbf{k}_j).$$

We experiment with both softmax and sigmoid operations for computing the attention weights in our hyperbolic models. The motivation for considering sigmoid attention weights is that in some applications (e.g. visual question answering), it makes sense for the attention weights over different entities to not compete with each other.

## 4 HYPERBOLIC ATTENTION NETWORKS

In this section we show how to redefine the attentive read operation of Section 3.1 as an operation on points in hyperbolic space. The key for doing this is to define new matching and aggregation functions that operate on hyperbolic points and take advantage of the metric structure of the manifold they live on. However, in order to apply these operations inside of a network we first we need a way to interpret network activations as points in hyperbolic space.

We describe how to map an arbitrary point in $\mathbb{R}^n$ onto the hyperboloid, where we can interpret the result as a point in hyperbolic space. The choice of mapping is important since we must ensure that the rapid scaling behavior of hyperbolic space is maintained. Armed with an appropriate mapping we proceed to describe the hyperbolic matching and aggregation operations that operate on these points.

### 4.1 HYPERBOLIC NETWORK ACTIVATIONS

Mapping neural network activations into hyperbolic space requires care, since network activations might live anywhere in $\mathbb{R}^n$, but hyperbolic structure can only be imposed on special subsets of Euclidean space (Krioukov et al., 2010). This means we need a way to map activations into an appropriate manifold. We choose to map into the hyperboloid, which is convenient since it is the only unbounded model of hyperbolic space in common use.

**Pseudo-polar coordinates:** In polar coordinates, we express an $n$-dimensional point as a scalar radius, and $n-1$ angles. Pseudo-polar coordinates consist of a radius $r$, as in ordinary polar coordinates, and an $n$-dimensional vector $\mathbf{d}$ representing the direction of the point from the origin. In the following discussion we assume that the coordinates are normalized, i.e. that $\|\mathbf{d}\| = 1$.

If $(\mathbf{d}, r) \in \mathbb{R}^{n+1}$ are the activations of a layer in the network, we map them onto the hyperbolid in $\mathbb{R}^{n+1}$ using $\pi((\mathbf{d}, r)) = (\sinh(r)\mathbf{d}, \cosh(r))$, which increases the scale by an exponential factor.

It is easily verified that the resulting point lies in the hyperboloid, and to verify that we maintain the appropriate scaling properties we compute the distance between a point and the origin using this projection:

$$d_{\mathbb{H}}(\mathbf{0}, (\mathbf{d}, r)) = \operatorname{arccosh}(-\langle \pi(\mathbf{0}), \pi((\mathbf{d}, r)) \rangle_M) = r \ ,$$

which shows that this projection preserves exponential growth in volume for a linear increase in $r$. Without the exponential scaling factor the effective distance of $\pi((\mathbf{d}, r))$ from the origin grows logarithmically in hyperbolic space.[1]

### 4.2 HYPERBOLIC ATTENTION

In this section, we show how to build an attentive read operation that operates on points in hyperbolic space. We consider how to exploit hyperbolic geometry in both the *matching* and the *aggregation* steps of the attentive read operation separately.

**Hyperbolic matching:** The most natural way to exploit hyperbolic geometry for matching pairs of points is to use the hyperbolic distance between them. Given a query $\mathbf{q}_i$ and a key $\mathbf{k}_j$ both lying in hyperbolic space we take,

$$\alpha(\mathbf{q}_i, \mathbf{k}_j) = f(-\beta d_{\mathbb{H}}(\mathbf{q}_i, \mathbf{k}_j) - c) \ , \tag{2}$$

where $d_{\mathbb{H}}(\cdot, \cdot)$ is the hyperbolic distance, and $\beta$ and $c$ are parameters that can be set manually or learned along with the rest of the network. Having the bias parameter $c$ is useful because distances are non-negative. We take the function $f(\cdot)$ to be either $\exp(\cdot)$, in which case we set the normalization appropriately to obtain a softmax, or $\operatorname{sigmoid}(\cdot)$.

---

[1] Alternatively we can treat $\mathbf{d}$ as a vector in the tangent space of $\mathbb{H}^n$ at the origin defining a geodesic and compute distances between points using the law of cosines, this leads to similar scaling properties.

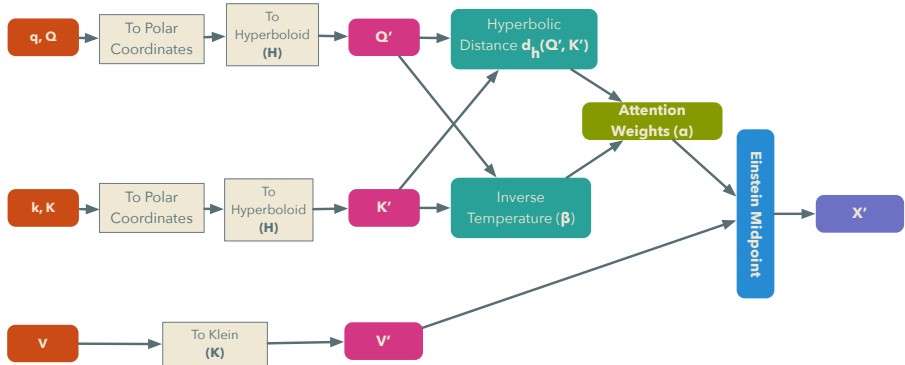

Figure 2: The computational graph for the self-attention mechanism of the hyperbolic Transformer.

**Hyperbolic aggregation:** The path to extend the weighted midpoint to hyperbolic space is much less obvious, but fortunately such a extension already exists as the Einstein midpoint. The Einstein midpoint is straightforward to compute by adjusting the aggregation weights appropriately (see Ungar (2005, Definition 4.21))

$$m_i(\{\alpha_{ij}\}_j, \{\mathbf{v}_{ij}\}_j) = \sum_j \left[ \frac{\alpha_{ij}\gamma(\mathbf{v}_{ij})}{\sum_\ell \alpha_{i\ell}\gamma(\mathbf{v}_{i\ell})} \right] \mathbf{v}_{ij} \quad , \tag{3}$$

where the $\gamma(\mathbf{v}_{ij})$ are the Lorentz factors, that are given by $\gamma(\mathbf{v}_{ij}) = \frac{1}{\sqrt{1-\|\mathbf{v}_{ij}\|^2}}$. The norm in the denominator of the Lorentz factor is the Euclidean norm of the Klein coordinates of the point $\mathbf{v}_{ij}$, and the correctness of Equation 3 also relies on the points $\mathbf{v}_{ij}$ being represented by their Klein coordinates. Fortunately the various models of hyperbolic space in common use are all isomorphic, so we can work in an arbitrary hyperbolic model and simply project to and from the Klein model to execute midpoint computations, as discussed in Section 2.

The reason for using the Einstein midpoint for hyperbolic aggregation is that it obeys many of the properties that we expect from a weighted average in Euclidean space. In particular, translating the $\mathbf{v}_{ij}$'s by a fixed distance in a common direction also translates the midpoint, and it is invariant to rotations of the constellation of points about the midpoint. The derivation of this operation is quite involved, and beyond the scope of this paper. We point the interested reader to Ungar (2005; 2008) for a full exposition.

## 5 EXPERIMENTS

We evaluate our models on synthetic and real-world tasks. Experiments where the underlying graph structure is explicitly known clearly show the benefits of using hyperbolic geometry as an inductive bias. At the same time, we show that real-world tasks within implicit graph structure such as a diagnostic visual question answering task (Johnson et al., 2017), and neural machine translation, equally benefit from relying on hyperbolic geometry. We provide experiments with feed-forward networks, the Transformer (Vaswani et al., 2017) and Relation Networks (Santoro et al., 2017) endowed with hyperbolic attention.

Our results show the effectiveness of our approach on diverse tasks and architectures. The benefit of our approach is particularly prominent in relatively small models, which supports our hypothesis that hyperbolic geometry induces compact representations and is therefore better able to represent complex functions in limited space.

### 5.1 MODELING SCALE-FREE GRAPHS

We use the algorithm of von Looz et al. (2015) to efficiently generate large scale-free graphs, and define two predictive tasks that test our model's ability to represent different aspects of the structure of these networks. For both tasks in this section, we train Recursive Transformer (RT) models, using hyperbolic and Euclidean attention. A Recursive Transformer is identical to the original transformer, except that the weights of each self-attention layer are tied across depth. Simultaneously to our work,

Figure 3: **Left:** Performance of the Recursive Transformer models on the Shortest Path Length Prediction task on graphs of various sizes. The black dashed line indicates chance performance. **Center:** Results on Link Prediction Tasks. **Right:** The histogram of the radiuses for a model trained on a graph with 100 and 400 nodes.

Dehghani et al. (2018) have proposed the same model as a generalization of the Transformer model and they referred to it as "Universal Transformers". We use models with 3 recursive self-attention layers, each of which has 4 heads with 4 units each for each of $\mathbf{q}$, $\mathbf{k}$, and $\mathbf{v}$. This model has similarities to Graph Attention Networks (Veličković et al., 2017; Kool and Welling, 2018).

**Link prediction (LP):** Link prediction is a classical graph problem, where the task is to predict if an edge exists between two nodes in the graph. We experimented with graphs of 1000 and 1200 nodes and observed that the hyperbolic RT performs better than the Euclidean RT on both tasks. We report the results in Figure 3 (middle). In general, we observed that for graphs of size 1000 and 1200 the hyperbolic transformer performs better than the Euclidean transformer given the same amount of capacity.

**Shortest path length prediction (SPLP):** In this task, the goal is to predict the length of the shortest path between a pair of nodes in the graph. We treat this as a classification problem with a maximum path-length of 25 which becomes naturally an unbalanced classification problem. We use rejection sampling during training to ensure the network is trained on an approximately uniform distribution of path lengths. At test time we sample paths uniformly at random, so the length distribution follows that of the underlying graphs. We report the results in Figure 3 (left). In Figure 3 (right), we visualize the distribution of the scale of the learned activations ($r$ in the projection of Section 4.1) when training on graphs of size 100 and 400. We observe that our model tends to use larger scales for the larger graphs. As a baseline, we compare to the optimal constant predictor, which always predicts the most common expected path length. This baseline does quite well since the path length distribution on the test set is quite skewed.

For both tasks, we generate training data online. Each example is a new graph in which we query the connectivity of a randomly chosen pair of nodes. To make training easier, we use a curriculum, whereby we start training on smaller graphs and gradually increase the number of vertices towards the final number. More details on the dataset generation procedure and the curriculum scheme are found in the supplementary material.

## 5.2 SORT-OF-CLEVR

Since we expect hyperbolic attention to be particularly well suited to relational modelling, we investigate our models on the relational variant of the Sort-of-CLEVR dataset (Santoro et al., 2017). This dataset consists of simple visual scenes allowing us to solely focus on the relational aspect of the problem. Our models extend Relation Nets (RNs) with the attention mechanism in hyperbolic space (with the Euclidean or Einstein midpoint aggregation), but otherwise we follow the standard setup-up (Santoro et al., 2017). Our best method yields accuracy of 99.2% that significantly exceeds the accuracy of the original RN (96%).

However, we are more interested in evaluating models on the low-capacity regime. Indeed, as Figure 4 (left) shows, the attention mechanism computed in the hyperbolic space improves around 20 percent points over the standard RN, where all the models use only two units of the relational MLP.

## 5.3 EXPERIMENTS ON CITESEER AND CORA

We use two of the standard graph transduction benchmark datasets, Citeseer and Cora (Sen et al., 2008) and used the same experimental protocol defined in Veličković et al. (2017). We use *graph attention*

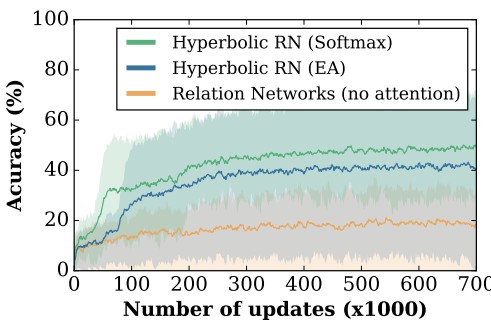 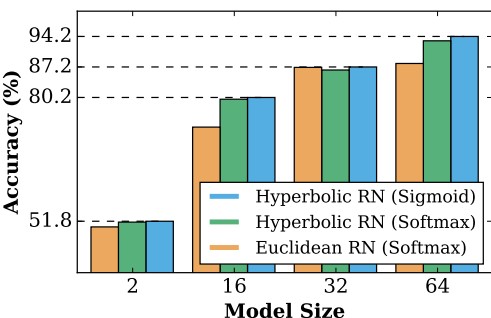

Figure 4: **Left:** Comparison of our models with low-capacity on the Sort-of-CLEVR dataset. The "EA" refers to the model that uses hyperbolic attention weights with Euclidean aggregation. **Right:** Performance of Relation Network extended by attention mechanism in either Euclidean or hyperbolic space on the CLEVR dataset.

| Transductive | | |
|---|---|---|
| **Method** | **Cora** | **Citeseer** |
| GCN (Kipf and Welling, 2016) | 81.5% | 70.3% |
| GAT (Veličković et al., 2017) | $83.0\% \pm 0.14$ | $72.5\% \pm 0.14$ |
| H-GAT | $\textbf{83.5\%} \pm \textbf{0.12}$ | $\textbf{72.9\%} \pm \textbf{0.078}$ |

Table 1: Results on graph transduction tasks. We have used the same setup that is described in (Veličković et al., 2017). H-GAT refers to our graph attention network with hyperbolic attention mechanism. Table shows the mean performance over 100 random seeds, along with 95% confidence intervals for this estimate.

*networks* (GAT) as our baseline and developed a hyperbolic version of GAT (H-GAT) by replacing the original attention mechanism with the hyperbolic attention using softmax.

We report our results in Table 1 and compare against the GAT with the Euclidean attention mechanism. We compute the standard deviations over 100 seeds and got improvements both on Citeseer and Cora datasets over the original GAT model. We show the visualizations of the learned hyperbolic embeddings of $\mathbf{q}$ and $\mathbf{k}$ in A.4.

## 5.4 CLEVR

We train our Relation Network with various attention mechanisms on the CLEVR dataset (Johnson et al., 2017). CLEVR is a synthetic visual question answering datasets consisting of 3D rendered objects like spheres, cubes, or cylinders of various size, material, or color. In contrast to other visual question answering datasets (Antol et al., 2015; Malinowski and Fritz, 2014; Zhu et al., 2016), the focus of CLEVR is on relational reasoning.

In our experiments, we closely follow the procedure established in (Santoro et al., 2017), both in terms of the model architecture, capacity, or the choice of the hyperparameters, and only differ by the attention mechanism (Euclidean or hyperbolic attention), or sigmoid activations.

Results are shown in Figure 4 (Right). For each model, we vary the capacity of the relational part of the network and report the resulting test accuracy. We find that hyperbolic attention with sigmoid consistently outperforms other models. Our RN with hyperbolic attention and sigmoid achieves 95.7% accuracy on the test set at the same capacity level as RN, whereas the latter reportedly achieves 95.5% accuracy (Santoro et al., 2017).

## 5.5 Neural machine translation

The Transformer (Vaswani et al., 2017) is a recently introduced state of the art model for neural machine translation that relies heavily on attention as its core operation. As described in Section 3.3,

| | WMT 2014 En-De BLEU Scores | | |
|---|:---:|:---:|:---:|
| | **Tiny** | **Base** | **Big** |
| Transformer (Vaswani et al. (2017)) | - | 27.3 | 28.4 |
| Transformer (Latest) | 17.3 | 27.1 | - |
| Hyperbolic Transformer (+Sigmoid) | 17.5 | 27.4 | - |
| Hyperbolic Transformer (+Softmax, +Pseudo-Polar) | 17.9 | 27.4 | - |
| Hyperbolic Transformer (+Sigmoid, +Pseudo-Polar) | **18.6** | **27.9** | **28.52** |

Table 2: Results for the WMT14 English to German translation task. Results are computed following the procedure in Vaswani et al. (2017). Citations indicate results taken from the literature. *Latest* is the result of training a new model using an unmodified version of the same code where we added hyperbolic attention (we have observed that the exact performance of the transformer on this task varies as the Tensor2tensor codebase evolves).

we have extended the Transformer[2] by replacing its scaled dot-product attention operation with its hyperbolic counterpart. We evaluate all the models on the WMT14 En-De dataset (Bojar et al., 2014).

We train several versions of the Transformer model with hyperbolic attention. They use different coordinate systems (Cartesian or pseudo-polar), or different attention normalization functions (softmax or sigmoid). We consider three model sizes, referred to here as *tiny*, *base* and *big*. The tiny model has two layers of encoders and decoders, each with 128 units and 4 attention heads. The base model has 6 layers of encoders and decoders, each with 512 units and 8 attention heads. All hyperparameter configurations for the Euclidean versions of these models are available in the Tensor2tensor repository.

Results are shown in Table 2. We observe improvements over the Euclidean model by using hyperbolic attention, in particular when coupled with the sigmoid activation function for the attention weights. The improvements are more significant when the model capacity is restricted.

In addition, our best model (with sigmoid activation function and without pseudo-polar coordinates) using the *big* architecture from Tensor2tensor, achieves 28.52 BLEU score, whereas Vaswani et al. (2017) report 28.4 BLEU score with the original version of this model.[3].

## 6 Conclusion

We have presented a novel way to impose the inductive biases from hyperbolic geometry on the activations of deep neural networks. Our proposed hyperbolic attention operation makes use of hyperbolic geometry in both the computation of the attention weights, and in the aggregation operation over values. We implemented our proposed hyperbolic attention mechanism in both Relation Networks and the Transformer and showed that we achieve improved performance on a diverse set of tasks. We have shown improved performance on link prediction and shortest path length prediction in scale free graphs, on two visual question answering datasets, real-world graph transduction tasks and finally on English to German machine translation. The gains are particularly prominent in relatively small models, which confirms our hypothesis that hyperbolic geometry induces more compact representations.

Yang and Rush (2016) have proposed to imposed the activations of the neural network to lie on a Lie-group manifold in the memory. Similarly as a future work, an interesting potential future direction is to use hyperbolic geometry as an inductive bias for the activation of neural networks in the memory.

## Acknowledgements

We would like to thank Neil Rabinowitz, Chris Dyer for constructive comments on earlier versions of this draft. We thank Yannis Asseal for helping us with the styles of the plots in this draft. We would like to thank Thomas Paine for the constructive comments.

---

[2]We use a publicly available version: `https://github.com/tensorflow/tensor2tensor`
[3]We achieve 28.3 BLEU score using the *big* Transformer with the publicly available framework.

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

# A APPENDIX

## A.1 MORE ON MODELS OF HYPERBOLIC SPACE

In Figure 5, we illustrate the relationship between different models of hyperbolic space. There are one-to-one isometric transformations defined between each different models of the hyperbolic space. Hyperboloid model is unbounded, whereas Klein and Poincare models are bounded in a disk.

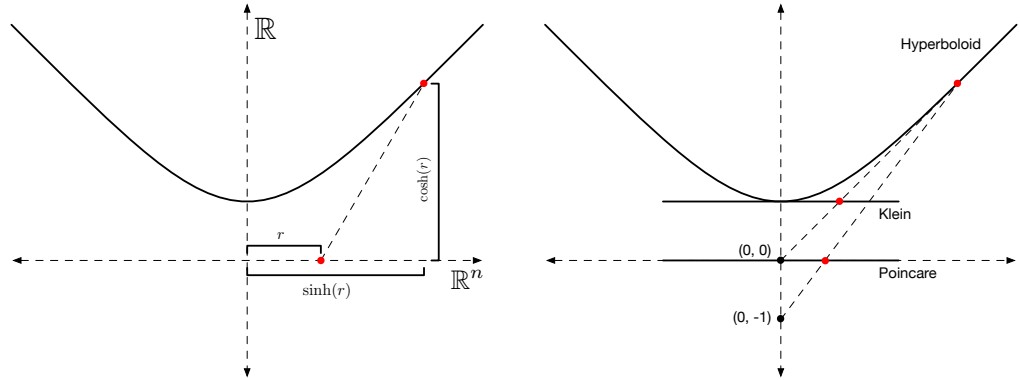

Figure 5: Relationships between different representations of points used in the paper. **Left:** The relationship between pseudo-polar coordinates in $\mathbb{R}^n$ and the hyperboloid in $\mathbb{R}^{n+1}$. **Right:** Projections relating the hyperboloid, Klein and Poincaré models of hyperbolic space.

## A.2 SCALE-FREE GRAPH GENERATION

We use the algorithm described by von Looz et al. (2015). In our experiments, we set the $\alpha$ to 0.95 and *edge_radius_R_factor* to 0.35. We will release our code both for generating and the operations in the hyperbolic space along with the camera-ready version of our paper.

## A.3 SCALE-FREE GRAPH CURRICULUM

Curriculum was an essential part of our training on the scale-free graph tasks. On LP and SPLP tasks, we use a curriculum where we extract the connected components from the graph by cutting the disk that the graphs generated on into slices by starting from a 30 degree angle and gradually increasing the size of the slice from the disk by increasing the angle during the training according to the number of lessons that are involved in the curriculum. This process is also visualized in Figure 6.

## A.4 VISUALIZATION OF QUERY AND KEY EMBEDDINGS ON CORA

In Figure 8 and 9, we visualize the embeddings of the query ($q$) and the keys ($k$) going into the hyperbolic matching function on the Poincare Ball model. In Figure 8, the embeddings of a model trained with dropout are bounded in a ball with smaller volume than the model trained without dropout. Also as clearly can be seen from the embedding visualizations **k**'s and **q**'s are clustered on different regions of the space.

## A.5 TRAVELLING SALESMAN PROBLEM (TSP)

We train an off-policy DQN-like agent (Mnih et al., 2015) with the HRT. The graphs for the TSP is generated following the procedure introduced in (Vinyals et al., 2015).

On this task, as an ablation we just compared the hyperbolic networks with and The results are provided in Figure 4 (Right) with and without implicit coordinates. Overall, we found that the hyperbolic transformer networks performs better when using the implicit polar coordinates.

## A.6 HYPERBOLIC RECURSIVE TRANSFORMER

As shown in Figure **??**, the hyperbolic RT is an extension of transformer that ties the parameters of the self-attention layers. The self-attention layer gets the representations of the nodes of the graph coming from the encoder and the decoder decodes that representation from the recursive self-attention layers for the prediction.

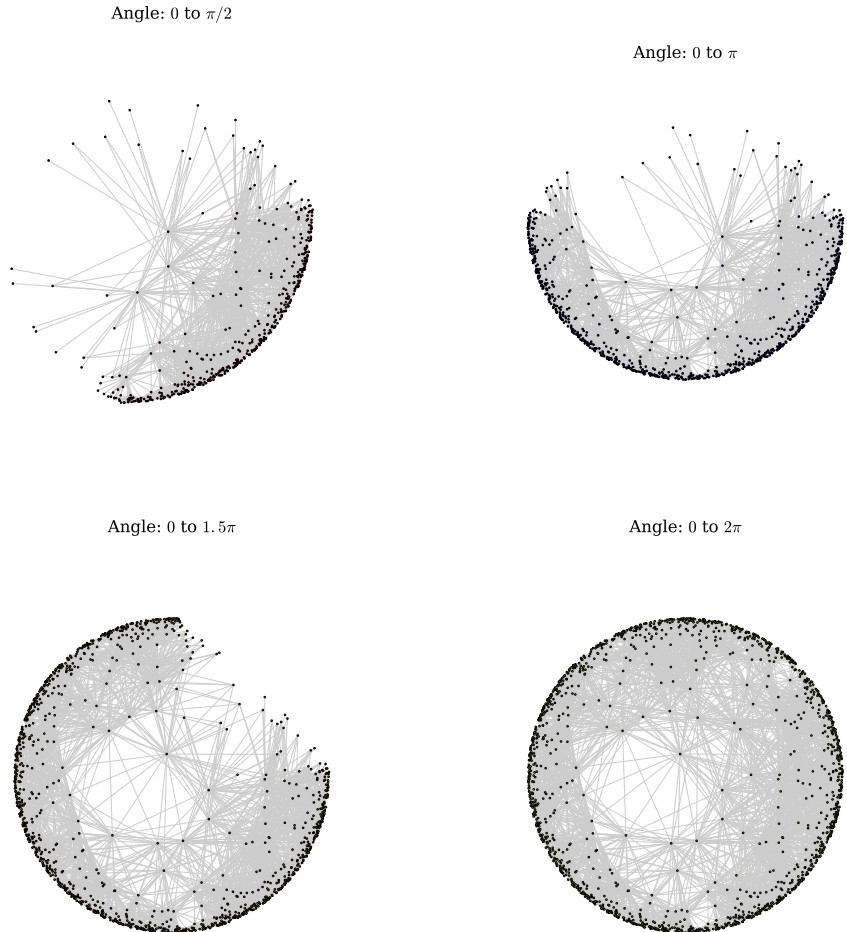

Figure 6: We show an example of a curriculum on the hyperbolic disk. In the first lesson, we take slices from the graph only between angle $0$ and $\pi/2$. In the second lesson we will have to take the slice from $0$ to $\pi$.

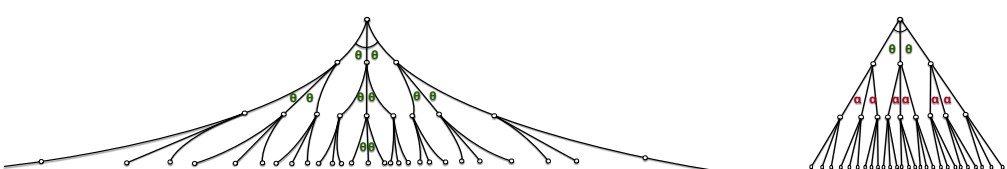

Figure 7: An illustration of how trees can be represented in hyperbolic (left) and Euclidean geometry (right) in a cone. In hyperbolic space, as the tree grows the angles between the edges ($\theta$) can be preserved from one level to the next. In Euclidean space, since the number of nodes in the tree grows faster than the rate that the volume grows, angles may not be preserved ($\theta$ to $\alpha$). Lines in the left diagram are straight in hyperbolic space, but appear curved in this Euclidean diagram.

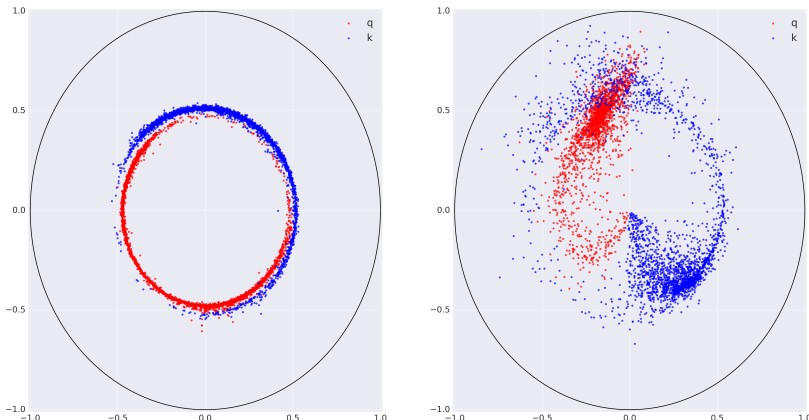

Figure 8: Hyperbolic embedding of $q$ (red) and $k$ (blue) in a Poincare Ball on Cora dataset. Each point corresponds to a node in the graph. This visualization is obtained from a model trained with dropout. The graph on the left is the embeddings going into the attention obtained from the first layer. The Figure on the right is for the embedding of the second layer.

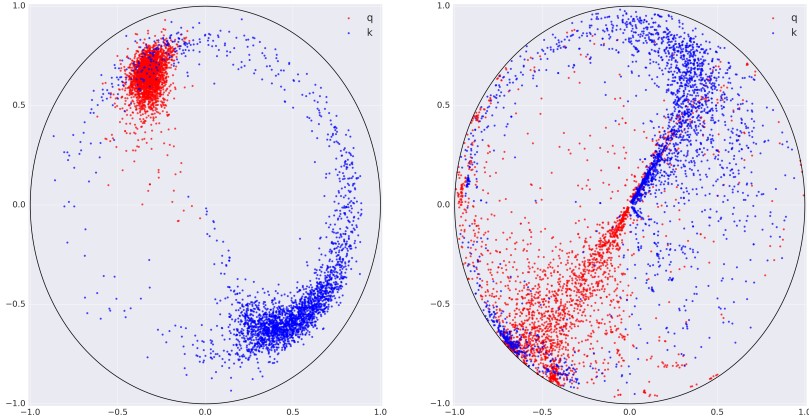

Figure 9: Hyperbolic embeddings of $\mathbf{q}$ (red) and $\mathbf{k}$ (blue) in a Poincare Ball on Cora dataset. Each point corresponds to a node in the graph. This visualization is obtained from a model trained without dropout. The figure on the left shows the embeddings going into the attention obtained from the first layer. The figure on the right shows the embedding of the second layer.

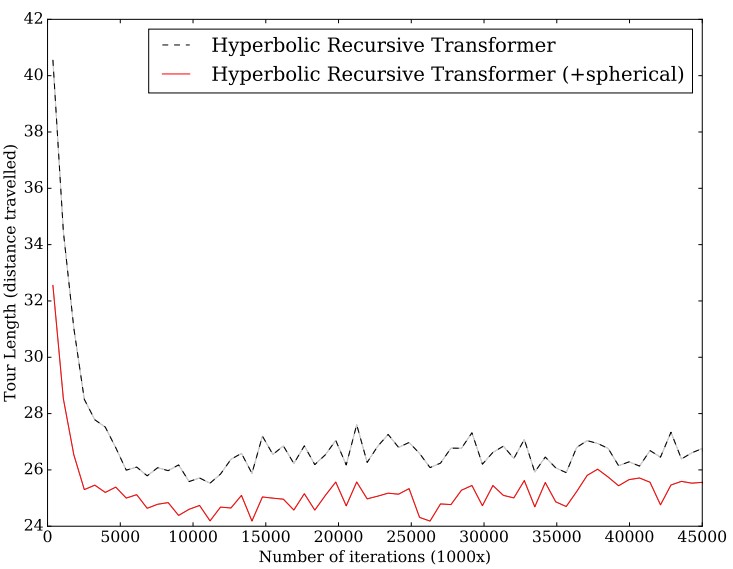

Figure 10: The comparisons between a hyperbolic recursive transformer with and without pseudo-polar (denoted as +spherical in the legend) coordinates on the travelling salesman problem.

