# OpenReview forum: "Hyperbolic Attention Networks"
_ICLR.cc/2019/Conference_

### Official Review · AnonReviewer2 · 2018-10-30
**ACCEPTABLE**

**Rating:** 7
**Confidence:** 4

**Review:**


The authors proposed to exploit hyperbolic geometry in computing the attention mechanisms for neural networks. Specifically, they break the attention read operation into two parts: matching and aggregation. In matching step, they use the hyperbolic distance to quantify the macthing between a query and a key; in the aggregation step, they use the Einstein midpoint. Their experiments results based on synthetic and real-world data shows the new method outperforms the traditional method based on Euclidean distance. This paper is acceptable.


Question: In Figure 3(Center), the number of nodes 1000 and 1200 are pretty close. How about the results on 500 nodes and 2000 nodes? It seems the accuracy difference increases as the number of nodes increases. Is this true?

---

> ### Author Response · Authors · 2018-11-23
> **On the Experiments with the Synthetic Graph Datasets**
>
> Thank you for your remarkable feedback and comments about our paper.
>
> > Question: In Figure 3 (Center), the number of nodes 1000 and 1200 are pretty close. How about the results on 500 nodes and 2000 nodes? It seems the accuracy difference increases as the number of nodes increases. Is this true?
>
> The difference in performance between the hyperbolic and Euclidean models is negligible if the graph is small enough (e.g. 200 nodes) and we would expect this trend to continue for graphs of size 2000 or larger, with the gap between hyperbolic and Euclidean models growing as the size of the graph increases. This is mainly because learning on larger graphs would require more capacity. We will add the experiments on graphs with 2000 nodes or larger to the camera-ready version of the paper.

---

### Official Review · AnonReviewer1 · 2018-11-02
**Refreshing approach for matching and aggregating**

**Rating:** 7
**Confidence:** 5

**Review:**

The authors propose a novel approach to improve relational-attention by changing the matching and aggregation functions to use hyperbolic geometric. By doing so the network can exploit the metric structure the functions live on.  Method was evaluated and showed improvements over baselines on wide range of tasks including translation, graph learning, and visual question answering.

Pros:
* High quality paper.
* Hyperbolic matching function is novel and interesting.
* Even though the subject isn’t trivial, the intuition was described well.
* The evaluation is comprehensive on several relational related tasks.

Cons:
* Baselines: The authors main contribution is the matching and aggregation operator. It always feels like the multi-modal community is divided between VQA and CLEVR datasets, but there should be a lot in common between them. Specifically, what is called here the matching operator, had several variants in VQA, such as Multimodal Compact Bilinear Pooling by Fukui et al., or Multi-modal Factorized Bilinear Pooling by You et al. etc. I think the paper would benefit from adding other variants of matching functions.
* Datasets: I think the approach might work as well in VQA dataset, which I find more interesting than clever because of the real-world nature of it. You can plug it into methods like MFB, or as pairwise potentials in Structured Attentions by Zhu et al, or High-Order attention by Schwartz et al

Conclusion:
A better matching and aggregating operations are always important, it can potentially improve performance in many challenges. The proposed method is novel and interesting, therefore I will be happy to see this paper as part of ICLR.

---

> ### Author Response · Authors · 2018-11-23
> **On Experiments with other VQA Models and Datasets**
>
>
> We really appreciate your feedback and comments about our paper.
>
> > Baselines: The authors main contribution is the matching and aggregation operator. It always feels like the multi-modal community is divided between VQA and CLEVR datasets, but there should be a lot in common between them. Specifically, what is called here the matching operator, had several variants in VQA, such as Multimodal Compact Bilinear Pooling by Fukui et al., or Multi-modal Factorized Bilinear Pooling by You et al. etc. I think the paper would benefit from adding other variants of matching functions.
> Datasets: I think the approach might work as well in VQA dataset, which I find more interesting than clever because of the real-world nature of it. You can plug it into methods like MFB, or as pairwise potentials in Structured Attentions by Zhu et al, or High-Order attention by Schwartz et al.
>
> In our work, we show the results of our hyperbolic module on a wide variety of different problems -- NMT, CLEVR, and graph problems. We used CLEVR because it has many relational questions that we hypothesize may benefit from representing in hyperbolic space. In contrast, the real-world VQA consists of somewhat shorter, and non-relational questions. Moreover, most of the challenges in the VQA dataset are about better visual representation that, in our work, we would like to abstract that away. Additionally, existing highly-performing architectures on CLEVR such as RN, also build upon the relational inductive biases, and hence there is a direct link between our module and these works. Nonetheless, we agree with the reviewers that the highly-performing VQA architectures may also benefit from our module, which we leave as possible future work.

---

### Official Review · AnonReviewer3 · 2018-11-02
**Applying a new metric to attention mechanism, improves small models but not large ones, reasonable but not very strong experimental comparisons.**

**Rating:** 6
**Confidence:** 5

**Review:**

This paper replaces the dot-product similarity used in attention mechanisms with the negative hyperbolic distance, and applies this new attention to the existing Transformer model, graph attention networks (GAT), and Relation Networks (RN). Accordingly, they use Einstein midpoint to compute the aggregation weights of attention in hyperbolic space. The idea of using hyperbolic rather than Euclidean space is based on the assumption that the input embeddings (neural net activations) are on the hyperbolic manifold, which follows power-law distribution and can be seem as a smooth description of tree-like hierarchy of data points. This assumption might hold for small neural networks with relatively low dimensional output. One main reason why this paper adopts the hyperbolic space is that the volume of hyperbolic space grows exponentially with the increase of radius while that of Euclidean space grows only polynomially. Using hyperbolic distance can increase the capacity of networks and handle the complexity of data. Experiments on Transformer and relation network show that Transformer, GAT and RN with the new attention metric produce better performance than Euclidean distance.

Pros:

1. Comparing to the existing methods using representations for shallow models in hyperbolic geometry, this paper extends the idea to deep neural networks.
2. The proposed attention mechanism can be easily applied to many of existing networks to enhance their capacity.
3. The experiments show several interesting results: 1) hyperbolic recursive transformer (RT) is consistently superior to Euclidean RT across the tasks in this paper; 2) hyperbolic space substantially benefits the low-capacity networks (i.e., low-dimensionality hidden state); 3) Einstein midpoint is better than Euclidean aggregation in hyperbolic space; 4) using sigmoid rather than softmax to compute attention weight may achieve better effectiveness on some tasks for the reason that the attention weights over different entities ay do not compete with each other.


Cons:

1. The novelty of this paper is replacing the Euclidean metric with another existing metric, which has already been used in previous ML models. So the contribution is limited.
2. As explicitly claimed in the paper and also reflected by the experimental results (e.g., Transformer-Big in Table 2). The hyperbolic metric only brings noticeable improvement to small neural nets with limited compacity on relatively small datasets. When applied it to most SOTA models (which are usually large/deep/wide neural networks) on larger datasets, it loses the advantage. This fact might seriously limit the application of the proposed technique.
3. Small models are preferred for inference especially on edge devices. But model compression and knowledge distillation can make small models having similar performance as large models, which might be much better than directly training a small model with the proposed metric.
4. Although hyperbolic metric reflects the power-law distribution, which is a very natural assumption verified on many kinds of real data (social networks and physical statistics), I am not fully convinced that it still holds on an embedding space produced by a neural net (since attention are usually applied to the outputs of a neural net).
5. In the experiments, does the model with the proposed metric cost similar training/inference time comparing to the baselines? What is the trade-off between improvements and extra time costs? I notice that the results of the proposed attention in Table 2 are ~0.5% higher than the results from the earlier arXiv version of this paper. What is the reason for the improvements? Did you increase the training steps?

---

> ### Author Response · Authors · 2018-11-23
> **A Response to Improves small models but not large ones, reasonable but not very strong experimental comparisons**
>
>
> Thank you for your constructive feedback and comments about our paper.
>
> > The novelty of this paper is replacing the Euclidean metric with another existing metric, which has already been used in previous ML models. So the contribution is limited.
>
> We present a method to ensure that the activations of a neural network can be interpreted as points in hyperbolic space, which is not guaranteed apriori. We show that by use of hyperbolic geometry to compute the attention, it is possible to reason over relations and graphs more efficiently and  with more compact representations.
>
> We are among the first to adopt the inductive biases from hyperbolic geometry to improve the attention mechanisms of neural networks, and we have done so in a general and modular way that can be used directly in any existing attention based architecture.
>
> > As explicitly claimed in the paper and also reflected by the experimental results (e.g., Transformer-Big in Table 2). The hyperbolic metric only brings noticeable improvement to small neural nets with limited compacity on relatively small datasets. When applied it to most SOTA models (which are usually large/deep/wide neural networks) on larger datasets, it loses the advantage. This fact might seriously limit the application of the proposed technique.
>
> A comparison between network compression techniques and hyperbolic attention is an exciting direction for future work.  Especially since in principle they are orthogonal approaches, and perhaps combining them could be more effective than either in isolation.
>
> > Although hyperbolic metric reflects the power-law distribution, which is a very natural assumption verified on many kinds of real data (social networks and physical statistics), I am not fully convinced that it still holds on an embedding space produced by a neural net (since attention are usually applied to the outputs of a neural net).
>
> It is important to note that the use of hyperbolic geometry is not an assumption about how the activations behave, it is a structure that is imposed on the activations as a modelling choice.   We use hyperbolic geometry to provide power law structure as an inductive bias to the model, not as an assumption about how the activations would behave in the absence of this imposed structure. The improvements we have observed on the graph tasks also indicate this performance gain.
>
> > In the experiments, does the model with the proposed metric cost similar training/inference time comparing to the baselines? What is the trade-off between improvements and extra time costs?
>
>  Indeed the costs are similar. All the operations that we use are simple element-wise scalar operations. These operations have a negligible contribution to the total computational cost.
>
> > I notice that the results of the proposed attention in Table 2 are ~0.5% higher than the results from the earlier arXiv version of this paper. What is the reason for the improvements? Did you increase the training steps?
>
> We observed that mentioned improvements mainly after training our model longer. In the latest version we trained our models for as many steps as the original Transformer [1].  We made this change after speaking with the authors of “Attention is All You Need” paper, who suggested this as a way to improve the performance of the model. This accounts for the increased performance compared to the Arxiv version of the paper.
>
> [1] Vaswani, A., Shazeer, N., Parmar, N., Uszkoreit, J., Jones, L., Gomez, A. N., ... & Polosukhin, I. (2017). Attention is all you need. In Advances in Neural Information Processing Systems(pp. 5998-6008).

---

### Public Comment · (anonymous) · 2018-09-29
**Not have results of Pubmed and PPI**

I would like to ask you whether you are still working on these two datasets or not tending to compare with other models on the two datasets?

---

> ### Author Response · Authors · 2018-10-03
> **Results on Pubmed and PPI**
>
> Hi,
>
> Thank you for your interest in our paper.
>
> We have provided results on VQA (CLEVR and Sort-of-CLEVR), Neural Machine Translation (WMT'14 En-De), Graph Classification Tasks (synthetic with different sizes), finally on transductive graph tasks such as Citeseer and Cora tasks in our paper. We covered a wide range of possible tasks that an attention mechanism with different architectures can be applied to. We have provided both extensive analysis and promising results on the tasks that we explored in our paper. Our goal in this paper was to show the generality of our approach on a wide range of tasks.
>
> For the time being, we do not have any plans to provide more experiments on other datasets besides the ones that are already presented in the paper.
>
> Best,

---

### Public Comment · (anonymous) · 2018-11-11
**Reminiscent of "Lie Access Neural Turing Machines"**

Dear authors,

This is quite an interesting work, utilizing hyperbolic geometry for more efficient representation. It reminds me of a previous work "Lie Access Neural Turing Machines" that proposed to use general manifolds as the "index space" of memory items, which are attended to like in standard attention. Could you comment on the relation of your paper to that work?

G. Yang and A. Rush. Lie-Access Neural Turing Machines. https://openreview.net/forum?id=Byiy-Pqlx&noteId=Byiy-Pqlx

---

> ### Author Response · Authors · 2018-11-23
> **Relevant Work - We will cite this paper**
>
> > This is quite an interesting work, utilizing hyperbolic geometry for more efficient representation. It reminds me of a previous work "Lie Access Neural Turing Machines" that proposed to use general manifolds as the "index space" of memory items, which are attended to like in standard attention. Could you comment on the relation of your paper to that work?
>
> Thanks for pointing out this paper, we were not aware of it.
>
> It indeed seems to be related to our method but with important distinctions. Yang et al introduces a memory access mechanism for NTM model where key and values of the memory are parametrized on a differentiable Lie-group manifold. To compute the matching function Yang et al also uses the distances between the key and value. Both Lie Access NTMs (LA-NTM) and our paper are closely related to each other in the sense of introducing attention-mechanisms by using certain geometric tools. However, the goals and the motivation of each work are quite different. LA-NTM represents uses the manifold of Lie-group actions to be able to learn better access mechanisms for the memory. Our paper focuses on improving the learning of relations in the data by combining attention mechanism with the hyperbolic inductive biases. It would be interesting to combine both approaches.
>
>  We will cite this work and relate it to our method in our revised manuscript.

---

### Meta-Review · Area_Chair1 · 2018-12-15
**Strong and interesting paper**

**Confidence:** 5
**Recommendation:** Accept (Poster)

**Metareview:**

Reviewers all agree that this is a strong submission.
I also believe it is interesting that only by changing the geometry of embeddings, they can use the space more efficiently without increasing the number of parameters.